# Prognostic factors in clinical stage IIIA small cell lung cancer: An analysis of a population-based cancer registry in Taiwan

Sung-Chi Yu[1]☯, Jing-Yang Huang[2,3]☯, Ya-Fu Cheng[1], Ching-Yuan Cheng[1,4], Chang-Lun Huang[1], Wei-Heng Hu[1], Bing-Yen Wang[1,4]*

1 Division of Thoracic Surgery, Department of Surgery, Changhua Christian Hospital, Changhua, Taiwan, 2 Institute of Medicine, Chung Shan Medical University, Taichung, Taiwan, 3 Center for Health Data Science, Chung Shan Medical University Hospital, Taichung, Taiwan, 4 Department of Post-Baccalaureate Medicine, College of Medicine, National Chung Hsing University, Taichung, Taiwan

☯ These authors contributed equally to this work.
* 156283@cch.org.tw

**Data Availability Statement:** Data cannot be shared publicly because of Institutional Review Board of Changhua Christian Hospital restriction. Data are available from the Taiwan Cancer Registry

## Abstract

Lung cancer stands as the primary cause of cancer-related death across the globe. The standard therapeutic approach for lung cancer involves concurrent chemoradiotherapy, with consideration of prophylactic cranial irradiation for younger or well-performing patients. In this study, we aimed to investigate prognostic factors and the impacts of different treatment methods on overall survival for stage IIIA small cell lung cancer in Taiwan. We obtained data from the Taiwan Cancer Registry, which included clinical and pathology data of 579 stage IIIA small cell lung cancer patients from January 2010 to December 2018, for this retrospective study. The enrolled patients had data on age, sex, Charlson Comorbidity Index score, histologic grading, clinical T, clinical N, clinical stage, treatment modality, and overall survival time. We compared overall survival among different subgroups to assess the impacts of these prognostic factors. The five-year survival rate for all patients was 20.57%, with a median survival time of 15.79 months. The data suggest that Charlson Comorbidity Index score, histologic grade, and clinical stage subgroups did not reach statistically significant differences. During the multivariate analysis, age over 70 years, sex, and treatment method were determined to be statistically significant independent prognostic factors. Patients who underwent surgical intervention exhibited significantly better outcomes compared to those who did not undergo operation.. In conclusion, stage IIIA small cell lung cancer is a highly heterogeneous disease. Operation should be considered as one of the alternative treatments in stage IIIA Small cell lung cancer patients.

## Introduction

Lung cancer stands as the primary cause of cancer-related death worldwide [1]. Based on a pathology report, lung cancer can be classified as small cell lung cancer or non-small cell lung cancer. Small cell lung cancer accounts for approximately 15% of all lung cancer patients [2].

for researchers who meet the criteria for access to confidential data. The data underlying the results presented in the study are available from Taiwan Cancer Registry. The email address of Taiwan Cancer Registry is nhird@nhri.org.tw

**Funding:** The author(s) received no specific funding for this work.

**Competing interests:** NO authors have competing interests

Due to its rapid doubling time, high growth fraction, and early development of widespread metastases, lung cancer presents significant challenges.

Only about 30% of patients are diagnosed with limited-stage small cell lung cancer, while the rest are diagnosed with extensive-stage small cell lung cancer. The prognosis remains poor, with a median survival time of 15–20 months in extensive-stage small cell lung cancer [3]. The standard treatment recommended by the current National Comprehensive Cancer Network guidelines for limited-stage small cell lung cancer is concurrent chemoradiotherapy [4]. Patients under 70 years old who have good performance status and positive responses to chemoradiotherapy may consider prophylactic cranial irradiation. Patients with stage T1-2N0M0 small cell lung cancer are recommended to undergo operation.

The complex clinicopathology of lung cancer causes difficulties in determining a treatment strategy. Several prognostic factors influence the survival rate, including poor performance status, extensive-stage disease, weight loss, lactate dehydrogenase levels, sex, and age [4]. In our previous study, we analyzed the heterogeneity and current treatment strategies of stage IIIA non small cell lung cancer patients in Taiwan [5]. Due to a lack of comprehensive analysis regarding treatment strategies for stage IIIA lung cancer patients in Taiwan, we designed this study to analyze stage heterogeneity and current treatment strategies of stage IIIA small cell lung cancer this time.

The purpose of this study is to identify prognostic factors that can aid in choosing an adequate treatment strategy for clinical stage IIIA small cell lung cancer. For this study, we collected data from the Taiwan Cancer Registry to determine prognostic factors and assess the influence of different treatment methods on overall survival for small cell lung cancer patients in Taiwan.

## Materials and methods

### Database and study sample

The study was conducted in accordance with the Declaration of Helsinki, and approved by the Institutional Review Board of Changhua Christian Hospital, the Internal Review Board number was 230505 and date of approval was May 24, 2023 for studies involving humans and the patient consent had been waived.

The dataset used in this study was obtained from the Taiwan Cancer Registry, which is maintained by the national health insurance system in Taiwan. This registry covers the entire population of Taiwan, totaling 23 million people. It provides comprehensive clinical and diagnostic laboratory data, as well as treatment methods since 1979.

The sample period was January 2010 to December 2018. All patients underwent tissue diagnosis confirmation and were diagnosed as clinical stage of small cell lung cancer. The clinical sample was identified using site codes from the International Classification of Diseases, 10th revision. Initially, a total of 101,261 patients with lung cancer were identified, including those with unspecified malignant neoplasms, non-small cell carcinomas (such as adenocarcinoma, squamous cell carcinoma, large cell carcinoma), small cell carcinomas, sarcomas, and other specified malignant neoplasms. To ensure data quality and relevance, we excluded 22,994 patients who had incomplete long-form cancer registry data, patients with an index year later than December 2018, patients under the age of 18, patients with missing clinical stage data, and patients who were deceased before the index date. As a result, we enrolled a total of 579 patients diagnosed with stage IIIA small cell lung cancer for this study (Fig 1). Illustrates the patient selection process.

After obtain the data, we were unable to access to information which could identify individual participants during or after data collection. In terms of assessing the clinical stage, the

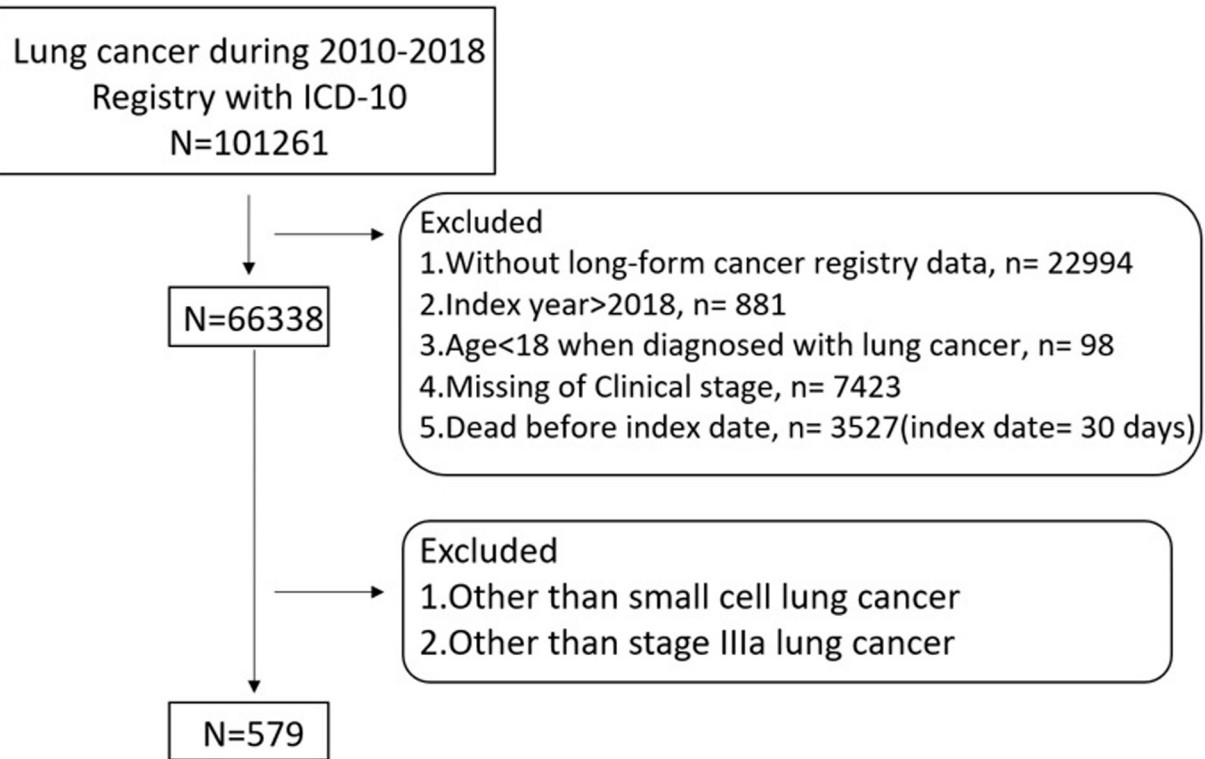

**Fig 1. Flowchart of patients enrollment process of study.**

National Health Insurance of Taiwan provided coverage for all preoperative staging evaluations. These evaluations included chest computed tomography scans, positron emission tomography/computed tomography scans with contrast, bronchoscopies, and brain magnetic resonance imaging. For mediastinal lymph node staging, endobronchial ultrasound-guided transbronchial needle aspiration was conducted for patients who tested positive on positron emission tomography/ Computed Tomography scans or had lymph nodes larger than 1 cm.

The enrolled patients' characteristics and demographic data were collected, including age, sex, Charlson Comorbidity Index score, histologic grading, clinical T stage, clinical N stage, clinical stage, treatment modality, and survival time. Tumor grading was based on the World Health Organization's classification system, and staging was performed according to the 7th edition of the tumor, node, metastasis (TNM) staging system. Our research protocol meet the guidelines of our Taiwan Society of Thoracic Surgeons specialists requirements of a specialist medical center.

## Outcome measures and analyses

The outcome measures for our study were based on the overall survival of small cell lung cancer patients. Survival time was defined as the period from confirmation of malignancy to either death or until December 2020. Survival curves were plotted using the Kaplan-Meier method and compared using the log-rank test.

Differences in survival estimates were calculated using the Cox proportional hazards regression model, which was stratified for hospital volume and adjusted for known prognostic factors. All statistical calculations were performed using the SAS, version 9.3, and Statistical Product and Service Solutions, version 20.

The overall survival rate was calculated using the Kaplan-Meier method. Univariate and multivariate analyses were conducted using the Cox proportional hazards model. The selected covariates, including age, sex, Charlson Comorbidity Index score, histology grade, clinical stage, and treatment modality, were associated with cancer prognosis. The univariate analysis included all selected clinicopathological factors. The background knowledge criteria [6] were applied for selecting all covariates in the multivariable analysis. A p-value of less than 0.05 was considered statistically significant.

## Results and discussion

The clinical and pathological characteristics of patients diagnosed with clinical stage IIIA small cell lung cancer were summarized in Table 1. Among the 579 small cell lung cancer patients in this study, more than four-fifths of them were male (n = 472).

Table 1. Patient demographic data and univariate survival analysis.

| Variable | Patients | 5-year survival rate, % (mean ± SE) | Median survival time, months (mean ± SE) | P-value |
|---|---|---|---|---|
| All | 579 | 20.57 ± 1.75 | 15.79 ± 0.73 | - |
| Age | | | | <0.0001 |
| 18–49 | 40 | 67.17 ± 7.48 | - | |
| 50–69 | 273 | 26.70 ± 2.79 | 21.81 ± 1.91 | |
| ≧70 | 266 | 7.53 ± 1.72 | 10.92 ± 6.80 | |
| Sex | | | | <0.0001 |
| Male | 472 | 16.60 ± 1.77 | 14.65 ± 0.70 | |
| Female | 107 | 38.50 ± 5.01 | 29.43 ± 4.32 | |
| CCI score | | | | 0.0830 |
| ≦2 | 361 | 21.78 ± 2.26 | 16.75 ± 0.95 | |
| 3–5 | 189 | 20.21 ± 3.04 | 14.92 ± 1.15 | |
| >5 | 29 | 8.28 ± 5.75 | 9.86 ± 2.31 | |
| Grade | | | | 0.1594 |
| Well | 13 | 41.67 ± 30.43 | 52.80 ±2.94 | |
| Moderate | 27 | 28.57 ± 17.07 | 27.00 ± 7.94 | |
| Poor | 208 | 21.59 ± 2.94 | 15.21 ± 1.55 | |
| Undifferentiated | 16 | 36.46 ± 12.29 | 18.00 ± 6.00 | |
| Unknown | 315 | 17.51 ± 2.16 | 15.52 ± 0.83 | |
| Clinical stage | | | | 0.8514 |
| T1N2 | 70 | 23.25 ± 5.21 | 16.91 ± 2.28 | |
| T2N2 | 182 | 22.17 ± 3.14 | 15.79 ± 1.07 | |
| T3N1 | 45 | 20.53 ± 6.51 | 19.00 ± 6.71 | |
| T3N2 | 163 | 20.22 ± 3.21 | 16.38 ± 1.60 | |
| T4N0 | 62 | 22.74 ± 5.50 | 12.00 ± 1.39 | |
| T4N1 | 57 | 9.85 ± 4.67 | 14.45 ± 2.06 | |
| Treatment | | | | < .0001 |
| None | 68 | 6.53 ± 3.20 | 10.80 ± 1.65 | |
| OP ± others | 62 | 50.09 ± 6.81 | 60.20 ± 14.07 | |
| CT ± others | 306 | 15.25 ± 2.12 | 15.53 ± 0.94 | |
| CRT | 43 | 37.34 ± 7.69 | 26.80 ± 18.33 | |
| Others | 100 | 21.79 ± 4.27 | 13.06 ± 1.76 | |

CCI = Charlson Comorbidity Index; OP: operation; CT: chemotherapy; CRT: chemoradiotherapy

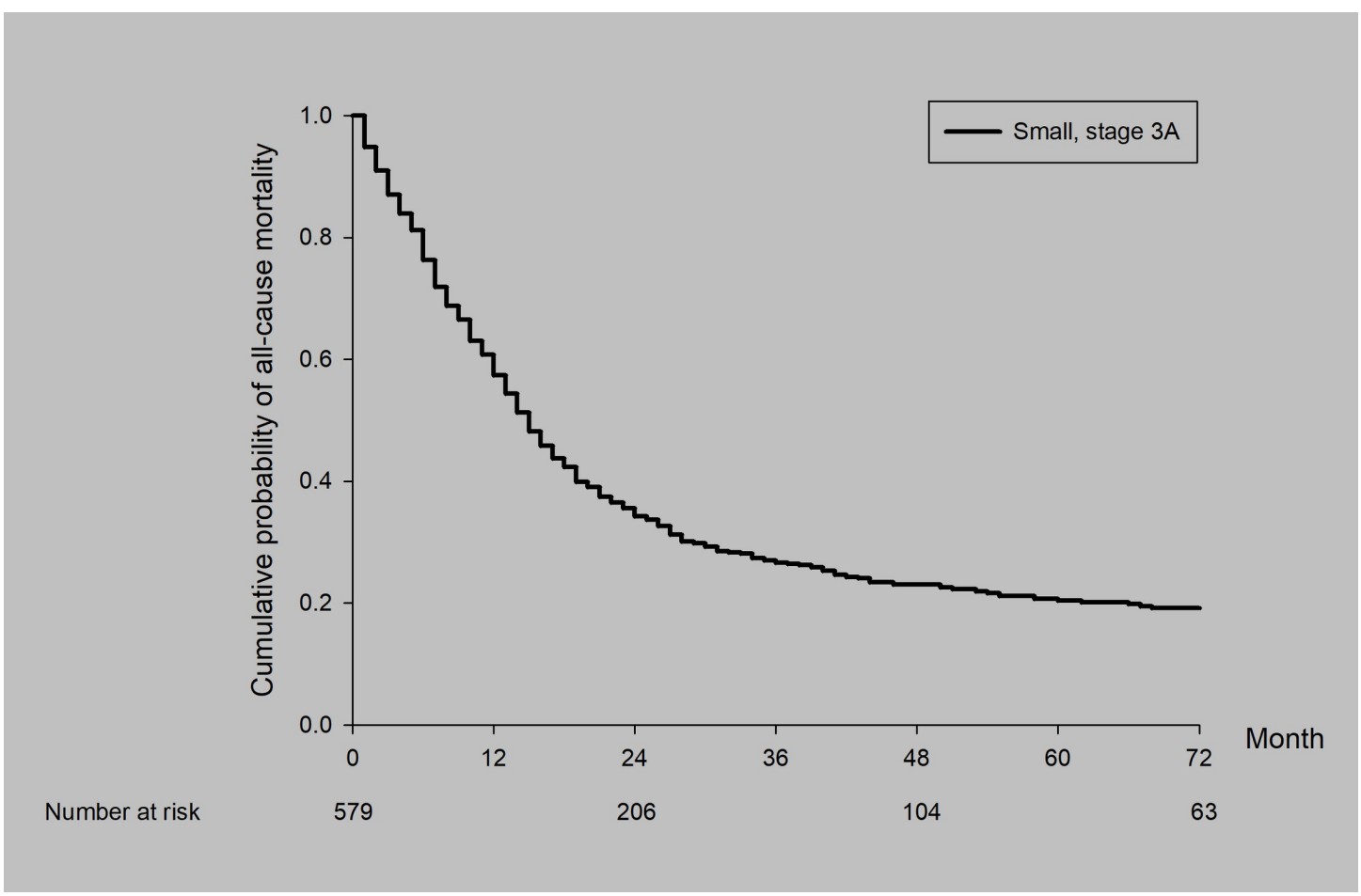

**Fig 2. Survival curves for all patients with stage IIIA SCLC.**

The 5-year survival rate was 20.6% with a median survival of 15.79 months (Fig 2). The 5-year survival rate and median survival time were assessed and stratified based on each clinical characteristic, including age, sex, Charlson Comorbidity Index score, histology grade, clinical stage, and treatment method.

The 5-year overall survival rates according to patient age were 67.2% (18–49 years), 26.7% (50–69 years), and 7.5% (>70 years). The differences in 5-year survival rate were significant (p<0.0001).

Patients were distributed according to clinical stage as follows: cT1N2 (n = 70, 12.0%), cT2N2 (n = 182, 31.4%), cT3N1 (n = 45, 7.8%), cT3N2 (n = 163, 28.2%), cT4N0 (n = 62, 10.7%), and cT4N1 (n = 57, 9.8%). The survival curves according to clinical stage are shown in (Fig 3). The 5-year survival rates by clinical stage were 23.3% for cT1N2, 22.2% for cT2N2, 20.5% for cT3N1, 20.2% for cT3N2, 22.7% for cT4N0, and 9.9% for cT4N1. There were no significant differences in survival rate among the clinical stages.

Male diagnosed with stage IIIA small cell lung cancer had poorer 5-year survival rates compared to female (16.6% vs. 38.5%), as shown in (Fig 4).

The 5-year survival rates according to histology were as follows: 41.7% for well differentiated (n = 13), 28.6% for moderately differentiated (n = 27), 21.6% for poorly differentiated

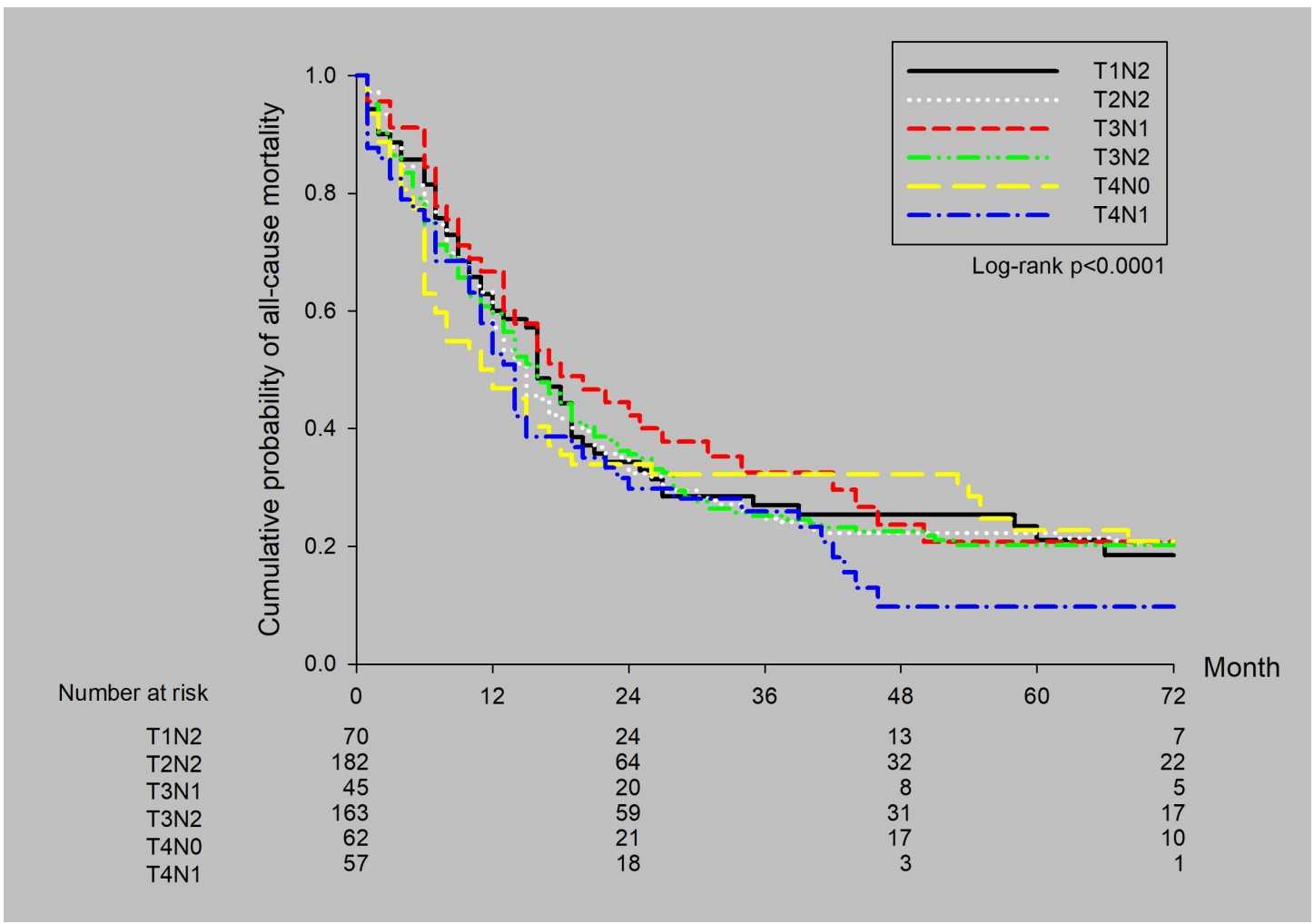

**Fig 3. Survival curves by clinical stage for all patients with stage IIIA SCLC.**

(n = 208), 36.5% for undifferentiated (n = 16), and 17.5% for unknown histology (n = 315). There were no significant differences in survival rate between the classes of differentiation.

Additionally, the 5-year survival rates according to Charlson Comorbidity Index score were 21.8% (Charlson Comorbidity Index score ≤2), 20.2% (Charlson Comorbidity Index score 3–5), and 8.3% (Charlson Comorbidity Index score >5). However, there was no significant difference between Charlson Comorbidity Index scores (p = 0.083).

Patients were distributed according to treatment method as follows: none (n = 68, 11.7%), operation (n = 62, 10.7%), chemotherapy (n = 306, 52.8%), chemoradiotherapy (n = 43, 7.4%), and other treatment (n = 100, 17.2%). The survival curves stratified by treatment method are shown in (Fig 5). The 5-year survival rates were 6.5% for the no treatment subgroup, 50.1% for the operation subgroup, 15.3% for the chemotherapy subgroup, 37.3% for the chemoradiotherapy subgroup, and 21.8% for the other treatment subgroup. There were significant differences between treatments.

In the univariate analysis, age, sex, and treatment method were found to be statistically associated with overall survival (Table 1). A multivariate Cox regression model was constructed, incorporating patient age, sex, Charlson Comorbidity Index score, histologic grade,

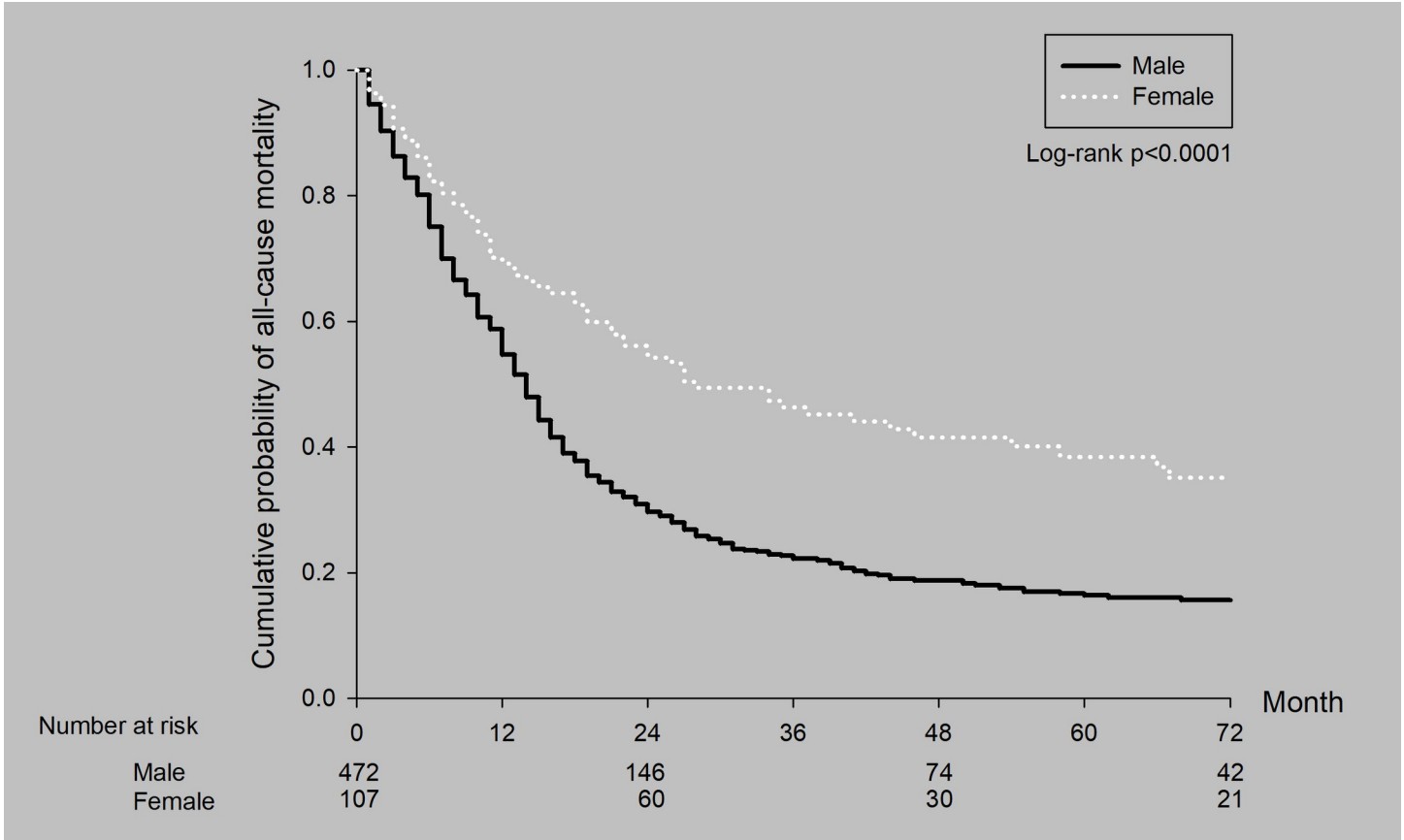

**Fig 4. Survival curves by sex for all patients with stage IIIA SCLC.**

and treatment method. The results of the multivariate analysis confirmed that age, sex, and treatment method were statistically significant independent prognostic factors for overall survival (Table 2).

## Discussion

This study aimed to investigate the clinical and pathological prognostic factors for patients with stage IIIA small cell lung cancer in Taiwan. We demonstrated that the different clinical factor of stage IIIA small cell lung cancer contributed to different outcomes. The results of our study indicated that age, sex, and treatment method were independent prognostic factors in the multivariate analysis. We also found that in all modality treatment group, the patient received operation revealed better 5-years survival rate than other patient did not received operative intervention.

Among the 579 patients diagnosed with stage IIIA small cell lung cancer, the impact of age on overall survival has been identified as an independent predictor in previous studies [7–9]. Lara et al. found that in patients with small cell lung cancer, age less than 50 years old was an independent predictor of improved cause-specific survival [7]. Wang et al. used the Surveillance, Epidemiology, and End Results population-based data to analyze clinicopathological characteristics of small cell lung cancer, especially in younger patients (age < 50) and older patients (age ≥ 50) [8]. They concluded that small cell lung cancer patients less than 50 years old had a better survival benefit, particularly for patients with American Joint Committee on

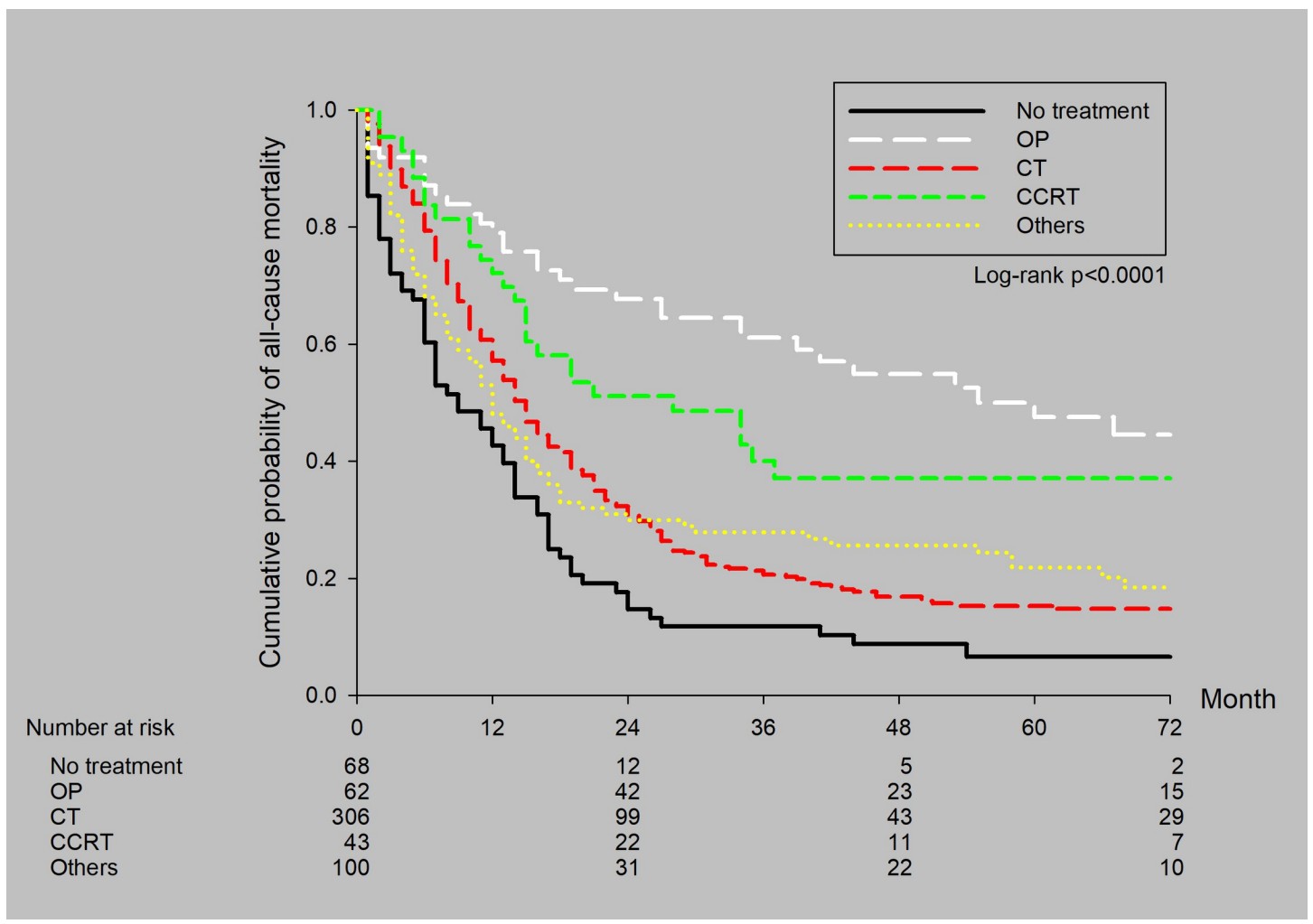

**Fig 5. Survival curves by treatment modality for all patients with stage IIIA SCLC.**

Cancer stage III small cell lung cancer. Our data revealed similar results. Patients who were 18–49 years old had a decreased risk of death compared to those 50–69 years old in the multivariate analysis. Age over 70 years at the time of diagnosis was an independent prognostic factor associated with an increased risk of death in our multivariate analysis. Previous studies have shown variations in clinicopathological features between younger and older patients [7–9]. Younger patients diagnosed with lung cancer are more likely to receive aggressive treatment and have better prognoses. However, older patients are less likely to undergo operation or radiation and have inferior outcomes compared to younger patients. Further studies are needed to understand the influence of age on survival.

Lung cancer appears to be more prevalent in male than female [1]. In our study, the male-to-female lung cancer incidence ratio was 4.41. This may be explained that male had a higher percentage of smokers in the Taiwan population. The female sex was identified as an independent prognostic factor for better survival in our multivariate analysis. The influence of sex on survival in patients with SCLC remains controversial [4, 10–14]. Ganti et al. indicated that females are associated with a more favorable prognosis in patients with limited-stage disease, but the female sex is not a favorable prognostic factor in patients with extensive-stage disease

**Table 2. Univariate and multivariate analyses of overall survival.**

| Variable | Univariate analysis | | | Multivariate analysis | | |
|---|---|---|---|---|---|---|
| | HR | 95% CI | P-value | HR | 95% CI | P-value |
| Age | | | | | | |
| 18–49 | 0.354 | (0.206–0.609) | 0.0002 | 0.370 | (0.213–0.643) | 0.0004 |
| 50–69 (reference) | 1 | | | 1 | | |
| ≧70 | 2.096 | (1.730–2.539) | < .0001 | 1.928 | (1.580–2.353) | < .0001 |
| Sex | | | | | | |
| Male (reference) | 1 | - | | 1 | | |
| Female | 0.566 | (0.436–0.736) | < .0001 | 0.659 | (0.503–0.864) | 0.0025 |
| CCI score | | | | | | |
| ≦2 (reference) | 1 | - | | 1 | | |
| 3–5 | 1.123 | (0.920–1.370) | 0.2542 | 1.103 | (0.900–1.351) | 0.3444 |
| >5 | 1.498 | (1.002–2.240) | 0.0491 | 1.069 | (0.705–1.621) | 0.7517 |
| Grade | | | | | | |
| Well (reference) | 1 | | | 1 | | |
| Moderate | 2.660 | (0.516–13.713) | 0.2423 | 2.241 | (0.426–11.773) | 0.3406 |
| Poor | 3.773 | (0.936–15.220) | 0.062 | 1.586 | (0.386–6.514) | 0.5224 |
| Undifferentiated | 2.857 | (0.639–12.767) | 0.1694 | 1.471 | (0.323–6.694) | 0.6172 |
| Unknown | 3.983 | (0.991–16.019) | 0.0516 | 1.496 | (0.364–6.144) | 0.5766 |
| Clinical stage | | | | | | |
| T1N2 (reference) | 1 | | | 1 | | |
| T2N2 | 1.054 | (0.773–1.438) | 0.7384 | 0.921 | (0.671–1.265) | 0.6114 |
| T3N1 | 0.944 | (0.613–1.453) | 0.7934 | 0.925 | (0.599–1.429) | 0.7249 |
| T3N2 | 1.012 | (0.738–1.386) | 0.9422 | 0.941 | (0.683–1.295) | 0.7073 |
| T4N0 | 1.010 | (0.682–1.495) | 0.9615 | 0.908 | (0.607–1.357) | 0.6375 |
| T4N1 | 1.223 | (0.818–1.827) | 0.3272 | 1.309 | (0.868–1.975) | 0.1993 |
| Treatment | | | | | | |
| None (reference) | 1 | | | 1 | | |
| OP ± others | 0.278 | (0.181–0.428) | < .0001 | 0.342 | (0.215–0.545) | < .0001 |
| CT ± others | 0.676 | (0.511–0.894) | 0.0061 | 0.715 | (0.535–0.955) | 0.0232 |
| CRT | 0.423 | (0.269–0.667) | 0.0002 | 0.528 | (0.332–0.839) | 0.0069 |
| Others | 0.654 | (0.466–0.916) | 0.0137 | 0.724 | (0.511–1.025) | 0.0684 |

HR = hazard ratio; CI: confidence interval; CCI = Charlson Comorbidity Index; OP: operation; CT: chemotherapy; CRT: chemoradiotherapy

[4]. However, Foster et al. indicated that sex was an important prognostic factor for both extensive-stage SCLC patients and limited-stage SCLC patients [14]. On the other hand, Lim et al. found no significant difference in survival based on sex in patients with resectable limited-stage SCLC [15]. Shan et al. conducted a study using the SEER database to establish a nomogram to predict the survival of SCLC patients with brain metastasis and reported that the female patients had a better outcome [11]. In our study, sex was still identified as an independent prognostic factor. The pathogenic features of SCLC in female may differ from those in male and require further investigation.

In our study, Charlson Comorbidity Index score was not identified as an independent prognostic factor. This finding is consistent with most studies that have shown no association between comorbidity and overall survival in small cell lung cancer patients [16–18]. In our study, the reason why better CCI score

did not provide better overall survival might be explained as follows. Elderly patients typically exhibit poor prognostic factors including poor performance status, impaired pulmonary function, or a combination of multiple comorbidities. In reverse, increased comorbidity does not always associate with these poor prognostic factors. Furthermore, patients with higher Charlson Comorbidity Index scores may still undergo concurrent chemoradiotherapy or even surgical interventions. Aarts et al. indicated that multimorbidity was associated with a slightly increased risk of death in patients with limited-stage SCLC, independent of treatment [19]. The treatment method and response may be influenced by comorbidity, but this could be attributed to increased age or poor performance status rather than comorbidity itself. Therefore, individualized treatment approaches should be considered for elderly patients with multimorbidity.

There have been few studies discussing the impact of histology grade on small cell lung cancer. Maeda et al. indicated that a poor histology grade was a significant independent risk factor for recurrence in patients with stage IIN0 and stage IIN1 non-small cell lung cancer [20]. Liu et al. also found that in patients with resected N1-stage II NSCLC, a poorly differentiated histological grade was a significant predictor of worse overall survival [21]. However, in our study, no histology tumor grade appeared to be an independent risk factor. Further studies are needed to investigate the effect of histology grade on small cell lung cancer patients.

In our study, the clinical stage of stage IIIA small cell lung cancer, including cT1N2, cT2N2, cT3N1, cT3N2, cT4N0, and cT4N1, was not identified as an independent prognostic factor. Lohinai et al. found that in patients with resected limited-stage small cell lung cancer, the clinical stage was not associated with overall survival [22]. Liu et al. also reported that in stage III SCLC patients who underwent chemotherapy or chest radiotherapy, the overall survival did not significantly differ between clinical stage IIIA and stage IIIB [23]. However, Wu et al. suggested that there is a lower risk of brain metastasis in stage I and stage II SCLC compared to stage III [24]. These findings suggest that patients with the same clinical stage may have similar overall survival, but further discussion is warranted for early or late-stage patients.

In our study, the treatment method emerged as the most valuable independent prognostic factor. Our research demonstrated that operation, chemotherapy, and chemoradiotherapy were associated with improved survival rates. This finding is consistent with previous studies that have also suggested the benefits of these treatments on overall survival [4, 25–28].

Among all the treatment methods, operation was the most effective predictor for improving overall survival in our study [29]. Over 70% of the operation were segmentectomies or lobectomies in stage III lung cancer group. The indications of segmentectomy in Taiwan might include poor pulmonary function or peripheral nodule less than 2 cm and no strong evidence of positive N2 lymph node stations. However, it is difficult to define the inoperability criteria precisely since the diversity of clinical condition and doctors' preferences. There might be selection bias due to most patients who received operation were highly selected which had the better prognosis then other stage IIIA SCLC patients. Besides, These patients were highly selected and they usually had less comorbidity and better performance status. Due to these patient are not quite common in stage IIIA SCLC. Further study focusing on the T3N1 stage comparing treatment strategy may need to conduct.

Recent studies also support the idea that patients with resectable limited SCLC have better outcomes [27, 30–33]. Takei et al. reported that age, sex, clinical stage, and surgical curability were independent prognostic factors, and surgical resection in patients with early-stage SCLC, especially stage I, resulted in better outcomes [31]. Su PL. reported that operation could be added as part of therapy for patient with stage III N2-positive NSCLC which could provide better progression-free survival. On the other hand, though the standard treatment for limited-stage SCLC is concurrent chemoradiotherapy, 306 patients received chemotherapy

instead of chemoradiotherapy due to reduced lung capacity, poor performance status, combination of multiple comorbidities and an inability to tolerate radiotherapy. In our study, we found that stage IIIA SCLC patient who followed the standard treatment strategy had better overall survival than those did not. We assume that operation may be considered an effective treatment option in selected patients with limited-stage SCLC, but further study need conduct to determined which factor was effective especially in clinical T3N1 stage patients.

We now consider the strengths and limitations of our study. One strength is that it is a population-based study using data obtained from the Taiwan Cancer Registry, which provides detailed information on surgical and pathological aspects such as laboratory data, chest and abdominal imaging, and treatment methods. This enabled us to include a large number of patients with small cell lung cancer in our study, providing valuable information for clinical treatment planning in patients with stage IIIA small cell lung cancer. However, our study also had several limitations. First, it was a retrospective study, which inherently introduces complexities, variations, and biases. Second, there may have been incomplete or inaccurate clinical pathology data during data collection. Furthermore, we have observed that younger age, lower Charlson Comorbidity Index, and clinically limited stage are associated with a tendency towards a better the 5-year survival rate. Therefore, patients with limited disease and better performance are likely selected to undergo operation. Furthermore, due to the high heterogeneity of the patient population, it may be challenging to generalize the conclusions of our study to all stage IIIA small cell lung cancer patients worldwide. In our study, operation should be considered as one of the alternative treatment in stage IIIA small cell lung cancer. The future research is required to explore the feasibility of operation in patients with stage IIIA lung cancer.

## Conclusion

In conclusion, the overall survival of stage IIIA small cell lung cancer patients in Taiwan remain poor. Our study identified age, sex, and treatment method as statistically significant independent prognostic factors. Analyzing these outcomes may help in developing more precise treatment strategies based on the patient's clinical condition and improve overall survival rates. It is important to consider comorbidity and age, as they can influence treatment methods and responses. These clinicopathological factors interact with each other, resulting in different treatment strategy plans and overall survival outcomes. Further investigations are needed to gain a better understanding of prognostic factors in small cell lung cancer and to refine treatment approaches in the pursuit of improved outcomes.

## Author Contributions

**Conceptualization:** Bing-Yen Wang.

**Data curation:** Ya-Fu Cheng, Ching-Yuan Cheng, Chang-Lun Huang, Wei-Heng Hu.

**Methodology:** Jing-Yang Huang.

**Software:** Jing-Yang Huang.

**Validation:** Jing-Yang Huang.

**Writing – original draft:** Sung-Chi Yu.

**Writing – review & editing:** Bing-Yen Wang.

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
