## [Decision Letter · Decision Letter 0]

28 Feb 2024

PONE-D-23-42734Prognostic factors in clinical stage IIIA small cell lung cancer: an analysis of a population-based cancer registry in TaiwanPLOS ONE

Dear Dr. Wang,

Thank you for submitting your manuscript to PLOS ONE. After careful consideration, we feel that it has merit but does not fully meet PLOS ONE’s publication criteria as it currently stands. Therefore, we invite you to submit a revised version of the manuscript that addresses the points raised during the review process.

We look forward to receiving your revised manuscript.

Kind regards,

Jun Hyeok Lim, M.D.

Academic Editor

PLOS ONE

Journal Requirements:

https://pubmed.ncbi.nlm.nih.gov/37493008/

In your revision ensure you cite all your sources (including your own works), and quote or rephrase any duplicated text outside the methods section. Further consideration is dependent on these concerns being addressed.

3. In the online submission form, you indicated that your data is available only on request from a third party. Please note that your Data Availability Statement is currently missing the contact details for the third party, such as an email address or a link to where data requests can be made. Please update your statement with the missing information. 

4. Please amend your manuscript to include your abstract after the title page.

Reviewers' comments:

Reviewer's Responses to Questions

**Comments to the Author**

1. Is the manuscript technically sound, and do the data support the conclusions?

Reviewer #1: Yes

Reviewer #2: Partly

Reviewer #3: Yes

2. Has the statistical analysis been performed appropriately and rigorously? 

Reviewer #1: I Don't Know

Reviewer #2: No

Reviewer #3: Yes

3. Have the authors made all data underlying the findings in their manuscript fully available?

Reviewer #1: Yes

Reviewer #2: No

Reviewer #3: Yes

4. Is the manuscript presented in an intelligible fashion and written in standard English?

Reviewer #1: Yes

Reviewer #2: No

Reviewer #3: Yes

5. Review Comments to the Author

Reviewer #1: Although the study potentially suggests that surgery is better for stage 3A small cell lung cancer, I have some concerns;

1- I think there is an important bias because those who are younger, have better performance, and have a more clinically limited disease are probably selected for the operation.

2- Whether there is a discrepancy between post-op pathology and pre-op clinical staging in patients who have undergone surgery seems to be a factor that may affect the study results

3- It is not stated in what proportion of patients PET-CT was used for staging, and whether bone scintigraphy was performed in addition to tomography for staging in those who did not use PET-CT. This raises questions about how many patients are accurately staged.

Despite all this, I see the study as an encouraging effort for surgery in selected patients with stage 3A small cell lung cancer.

Reviewer #2: In this current research, the authors aimed to investigate prognostic factors and the impacts of different treatment methods on overall survival for stage IIIa small cell lung cancer in Taiwan.

Comments:

1. Please try restructuring the introduction by dividing it into paragraphs.

2. "Variables with a p-value less than 0.10 in the univariate analysis were entered into the multivariate analysis,..."

Please give a reference for this sentence. Why did you include the .10 ones in the multivariable analysis?

3. In some places, the expression "men" and, in others, "male" were used. Please provide a standard.

4. Regarding the issue I mentioned in the second comment, it seems that multivariable analyses of all variables were made in the tables?

5. There is no need to give values such as p values and hazard ratios in the discussion.

6. Please do not use further abbreviations after stating the abbreviations. E.g. Charlson Comorbidity Index (CCI) score

7. Express the conclusion section more clearly. Why is future research needed? Also provide a perspective for future research.

8. There were lots of grammatical and spelling errors throughout the manuscript. The authors should meticulously revise their manuscript regarding typos and grammar. A native speaker's revisions would be beneficial.

Although the manual provides valuable information, it needs to be reviewed meticulously.

Reviewer #3: The authors are doing survival analysis based on clinical features. This paper is statistical analysis report of Taiwan Cancer Registry. Surgery and stages are important in prognosis and they validated that in new data set.

6. PLOS authors have the option to publish the peer review history of their article (what does this mean?). If published, this will include your full peer review and any attached files.

Reviewer #1: No

Reviewer #2: No

Reviewer #3: No

---

## [Author Response · Author response to Decision Letter 0]

27 Mar 2024

Reviewer #1: Although the study potentially suggests that surgery is better for stage 3A small cell lung cancer, I have some concerns;

1- I think there is an important bias because those who are younger, have better performance, and have a more clinically limited disease are probably selected for the operation

Thanks for your suggestion. Although the NCCN guidelines do not recommend

surgery as an standard treatment for stage IIIA small cell lung cancer. In our study, we introduced a new perspective that surgery should be considered as one of the alternative treatment. Furthermore, we have observed that young age, lower Charlson Comorbidity Index and clinically limited stage , the 5-year survival rate tends to be better(detailed data have been included in table. 1:

Patient demographic data and univariate survival analysis)

2- Whether there is a discrepancy between post-op pathology and pre-op clinical staging in patients who have undergone surgery seems to be a factor that may affect the study results

Thanks for your suggestion. We also consider that post-op pathology and pre-op clinical staging is an important factor may affect the 5-years overall. We will arrange further research that post-op pathology and pre-op clinical staging in patients whether affect the overall survival rate.

3- It is not stated in what proportion of patients PET-CT was used for staging, and whether bone scintigraphy was performed in addition to tomography for staging in those who did not use PET-CT. This raises questions about how many patients are accurately staged.

Despite all this, I see the study as an encouraging effort for surgery in selected patients with stage 3A small cell lung cancer..

Thanks for your suggestion. We also understand the importance of PET-CT and bone scan in accurately staged. The Taiwan National Health Insurance system in taiwan provides full coverage for PET-CT and bone scans. Therefore, overall, the clinical staging of lung cancer patients is relatively accurate.

Reviewer #2: In this current research, the authors aimed to investigate prognostic factors and the impacts of different treatment methods on overall survival for stage IIIa small cell lung cancer in Taiwan.

Comments:

1. Please try restructuring the introduction by dividing it into paragraphs.

Thanks for your suggestion, we will reorganize the introduction as requested, dividing it into paragraphs. 

2. "Variables with a p-value less than 0.10 in the univariate analysis were entered into the multivariate analysis,..."

Please give a reference for this sentence. Why did you include the .10 ones in the multivariable analysis?

Bursa et al. (Source Code Biol Med. 2008; 3: 17) established criteria for the purposeful selection of covariates, setting the variable entry threshold at 0.25 and the variable retention threshold at 0.10 to minimize discrepancies due to non-comparable parameters. The purposeful selection of covariates aims to minimize discrepancies arising from non-comparable parameters. However, we did not employ the purposeful selection of covariates. Instead, we revised the statistical statement "Variables with a p-value less than 0.10 in the univariate analysis were entered into the multivariate analysis..." to "The selected covariates, including age, sex, Charlson Comorbidity Index (CCI) score, histology grade, clinical stage, and treatment modality, were associated with cancer prognosis. The univariate analysis included all selected clinicopathological factors. The background knowledge criteria (refer to Biom J. 2018 May; 60(3): 431–449) were applied for selecting all covariates in the multivariable analysis."

3. In some places, the expression "men" and, in others, "male" were used. Please provide a standard.

Thanks for your valuable suggestion. We will correct the grammar and inconsistency in the article.

4. Regarding the issue I mentioned in the second comment, it seems that multivariable analyses of all variables were made in the tables?

We revised the statistical statement "Variables with a p-value less than 0.10 in the univariate analysis were entered into the multivariate analysis..." to "The selected covariates, including age, sex, Charlson Comorbidity Index (CCI) score, histology grade, clinical stage, and treatment modality, were associated with cancer prognosis. The univariate analysis included all selected clinicopathological factors. The background knowledge criteria (refer to Biom J. 2018 May; 60(3): 431–449) were applied for selecting all covariates in the multivariable analysis."

5. There is no need to give values such as p values and hazard ratios in the discussion.

Thanks for your valuable suggestion. We will remove any redundant descriptions to make the article more concise.

6. Please do not use further abbreviations after stating the abbreviations. E.g. Charlson Comorbidity Index (CCI) score

Thanks for your valuable suggestion. We will correct further abbreviations

7. Express the conclusion section more clearly. Why is future research needed? Also provide a perspective for future research.

Thanks for your valuable suggestion. In our study, surgery should be considered as one of the alternative treatment in stage IIIA small cell lung cancer. We will delete the below paragraph (Therefore, further investigation is required to gain more profound insights into these prognostic factors). The future research we will explore the feasibility of surgery in patients with stage IIIA lung cancer and include more relevant data. E.g. post-op pathology and pre-op clinical staging

8. There were lots of grammatical and spelling errors throughout the manuscript. The authors should meticulously revise their manuscript regarding typos and grammar. A native speaker's revisions would be beneficial.

Thanks for your valuable suggestion, we will correct any grammatical and spelling errors and seek revisions from a native speaker.

Although the manual provides valuable information, it needs to be reviewed meticulously.

Reviewer #3: The authors are doing survival analysis based on clinical features. This paper is statistical analysis report of Taiwan Cancer Registry. Surgery and stages are important in prognosis and they validated that in new data set.

Thank you for taking the time to review our paper. We are grateful to the reviewers for their valuable feedback and recommendations, which will help to improve the clarity and conciseness of our article.

---

## [Decision Letter · Decision Letter 1]

15 Jul 2024

Prognostic factors in clinical stage IIIA small cell lung cancer: an analysis of a population-based cancer registry in Taiwan

PONE-D-23-42734R1

Dear Dr. Wang,

We’re pleased to inform you that your manuscript has been judged scientifically suitable for publication and will be formally accepted for publication once it meets all outstanding technical requirements.

Kind regards,

Fumihiro Yamaguchi

Academic Editor

PLOS ONE

Additional Editor Comments (optional):

Reviewers' comments:

Reviewer's Responses to Questions

**Comments to the Author**

1. If the authors have adequately addressed your comments raised in a previous round of review and you feel that this manuscript is now acceptable for publication, you may indicate that here to bypass the “Comments to the Author” section, enter your conflict of interest statement in the “Confidential to Editor” section, and submit your "Accept" recommendation.

Reviewer #2: All comments have been addressed

Reviewer #4: All comments have been addressed

2. Is the manuscript technically sound, and do the data support the conclusions?

Reviewer #2: Yes

Reviewer #4: Yes

3. Has the statistical analysis been performed appropriately and rigorously? 

Reviewer #2: Yes

Reviewer #4: Yes

4. Have the authors made all data underlying the findings in their manuscript fully available?

Reviewer #2: Yes

Reviewer #4: Yes

5. Is the manuscript presented in an intelligible fashion and written in standard English?

Reviewer #2: Yes

Reviewer #4: Yes

6. Review Comments to the Author

Reviewer #2: The authors addressed the related comments mentioned. The rebuttals against the queries are satisfactory.

Reviewer #4: The authors touched on an important point. The current review is a well written review with relatively large patients numbers of a specific stage of small cell lung cancer. The authors have adequately addressed all the comments comments

7. PLOS authors have the option to publish the peer review history of their article (what does this mean?). If published, this will include your full peer review and any attached files.

Reviewer #2: No

Reviewer #4: **Yes: **Guler Yavas

---

## [Editor Report · Acceptance letter]

20 Jul 2024

PONE-D-23-42734R1 

PLOS ONE

Dear Dr. Wang, 

I'm pleased to inform you that your manuscript has been deemed suitable for publication in PLOS ONE. Congratulations! Your manuscript is now being handed over to our production team.

Kind regards, 

on behalf of

Dr. Fumihiro Yamaguchi 

Academic Editor

PLOS ONE